# Short-, Mid-, and Long-Term Epidemiological and Economic Effects of the World Bank Loan Project on Schistosomiasis Control in the People’s Republic of China

**DOI:** 10.3390/diseases10040084

**Published:** 2022-10-08

**Authors:** Qin Li, Jing Xu, Shi-Zhu Li, Jürg Utzinger, Donald P. McManus, Xiao-Nong Zhou

**Affiliations:** 1National Institute of Parasitic Diseases at Chinese Center for Disease Control and Prevention (Chinese Center for Tropical Diseases Research), NHC Key Laboratory of Parasite and Vector Biology, WHO Collaborating Centre for Tropical Diseases, National Center for International Research on Tropical Diseases, Shanghai 200025, China; 2School of Global Health, Chinese Center for Tropical Diseases Research, Shanghai Jiao Tong University School of Medicine, One Health Center, Shanghai Jiao Tong University-The University of Edinburgh, Shanghai 200025, China; 3Swiss Tropical and Public Health Institute, CH-4123 Allschwil, Switzerland; 4University of Basel, CH-4001 Basel, Switzerland; 5Molecular Parasitology Laboratory, QIMR Berghofer Medical Research Institute, Brisbane 4006, Australia

**Keywords:** control, economic evaluation, elimination, epidemiology, People’s Republic of China, schistosomiasis, World Bank Loan Project

## Abstract

It is widely acknowledged that the 10-year World Bank Loan Project (WBLP) on schistosomiasis control in the People’s Republic of China played an important role in raising the public and political profile of schistosomiasis, particularly regarding its prevention, control, and elimination. The WBLP adopted large-scale administration of praziquantel as the main control measure. At the end of the 10-year project in 2001, data from high-, medium-, and low-endemic areas suggested that the infection rates of both humans and domestic animals had fallen to the expected levels. However, major floods in the Yangtze River basin, coupled with reduced funding for schistosomiasis control, resulted in a rebound of the disease in endemic areas. Since 2005, a steady decline in infection rates was observed and it was hypothesized that the experiences and technological advances accumulated during the WBLP played a role. Nonetheless, relatively little is known about the long-term effects of the WBLP on schistosomiasis, particularly management mechanisms, technological innovations, epidemiological changes, and long-term economic impact. To fill these gaps, we systematically searched the literature for articles in English and Chinese on the WBLP on schistosomiasis from 1 January 1992 to 30 July 2022. Relevant studies were analyzed for short-, mid-, and long-term epidemiological and economic effects of the WBLP on schistosomiasis prevention, control, and elimination. Overall, 81 articles met our inclusion criteria, of which 17 were related to management mechanism reform, 20 pertained to technological innovation, and 44 examined epidemiological changes and economic effects. Most papers documented the WBLP as a positive contribution to schistosomiasis prevention and control in the People’s Republic of China. Regarding the long-term effects, there was a significant contribution to the national schistosomiasis control and elimination programme in terms of renewed management mechanisms, talent development, and technological innovation. In conclusion, the WBLP contributed to enhanced control of schistosomiasis and shaped the ultimate response towards schistosomiasis elimination in the People’s Republic of China. Experiences and lessons learned might guide schistosomiasis control and elimination elsewhere.

## 1. Introduction

Schistosomiasis remains a public health problem in economically disadvantaged regions of Africa, Asia, Latin America, and the Middle East [1,2]. Schistosomiasis japonica, due to infection with the zoonotic trematode *Schistosoma japonicum,* has been documented in the People’s Republic of China for more than 2100 years. In the early 1950s, there were an estimated 10 million people infected in 17 provinces [3]. The disease is less prevalent in economically favored areas. For instance, in the 1980s, the prevalence of schistosomiasis japonica in Japan was near zero [4]. In the People’s Republic of China, schistosomiasis is particularly prevalent in the Yangtze River valley, which is a complex geographical area. The disease also remains a concern in the Philippines [5].

In areas where schistosomiasis is endemic, sanitation is generally poor and severe cases of splenomegaly, ascites, colonic neoplasia, and growth retardation due to the lack or delayed treatment after infection are common [6,7,8]. In the mid-1950s, in view of the severe clinical presentation and the considerable public health implications of schistosomiasis, the Chinese government launched a national control programme. More than six decades of sustained efforts substantially improved the situation and reduced the burden of schistosomiasis and other neglected tropical diseases [9]. There is no doubt that, during this period, the World Bank Loan Project (WBLP) on schistosomiasis control in the People’s Republic of China played an important role. The WBLP was launched in 1992 in the provinces of Sichuan and Yunnan. Overall, the project covered 219 counties in eight provinces, namely Anhui, Hubei, Hunan, Jiangsu, Jiangxi, Sichuan, Yunnan, and Zhejiang. The project was completed in Anhui, Jiangsu, Jiangxi, Sichuan and Zhejiang by the end of 1998, while activities continued until 2001 in Hubei, Hunan, and Yunnan. The total financial investment in the WBLP was CNY 1088 million (USD 131.4 million, according to USD exchange rates in the year 2000) of which CNY 491 million was provided by the World Bank and the remaining CNY 567 million was matched by the Chinese government at all levels [10].

In high-schistosomiasis-endemic areas, the World Health Organization (WHO) recommends morbidity control. As control progresses, the focus shifts to infection control and, finally, to breaking transmission [5]. The stated objectives of the WBLP were to reduce infection rates in both humans and animals in endemic areas by 40% and to substantially reduce positive snail rates. Indeed, positive snail densities were reduced by 50–60% by the end of the WBLP, as the result of an integrated control approach, which placed particular emphasis on mass drug administration (MDA) for humans and cattle, improved information, education, and communication (IEC), complemented with focal snail control [11]. Of note, specific implementation mixes depended on *S. japonicum* infection prevalence in particular endemic areas. For instance, MDA was implemented in high endemicity areas, while selective chemotherapy was offered in low- and medium-endemic areas, complemented by snail control through environmental management.

The purpose of this report was to document short-, mid-, and long-term effects of the WBLP, placing particular emphasis on management mechanisms, technological innovations, epidemiological changes, and the resulting economic impact. We systematically reviewed the literature, synthesized and analyzed the recorded data, and draw experiences and lessons for schistosomiasis control and elimination in the People’s Republic of China.

We dedicate this paper to Professor Marcel Tanner in celebration of his 70th birthday. Our article complements another paper that pursued an elegant spatial analysis of schistosomiasis in the provinces of Hunan and Jiangxi [12]. Together, these two papers honour Tanner’s contribution to a deeper understanding of the epidemiology and control of schistosomiasis in the People’s Republic of China and elsewhere.

## 2. Materials and Methods

### 2.1. Search Strategy

We conducted a systematic literature review with an emphasis on the epidemiology and economic impact of the WBLP on schistosomiasis control in the People’s Republic of China. Particular attention was paid to management mechanisms, technological innovations, and changes in the epidemiological profiles during (1992–2001) and after the completion of the WBLP (2002–2022). We searched PubMed, Zhi Wang, Web of Science, and Science Direct from 1 January 1992 to 31 July 2022 for articles published in English and Chinese.

The following search terms and Boolean operators were employed: (i) “*Schistosoma*” OR “*Schistosoma japonicum*” OR “schistosomiasis japonica” OR “schistosomiasis”; (ii) “prevention and control” OR “primary prevention” OR “secondary prevention” OR “tertiary prevention”; (iii) “World Bank Loan Project” OR “WBLP”; (iv) “China” OR “People’s Republic of China”, and (v) “economic effect” OR “economic impact” OR “cost–benefit analysis” OR “cost-utility analysis” OR “cost-effectiveness analysis”. The bibliographies of relevant articles were hand-searched for identification of additional references. This search was continued until no new references were identified.

### 2.2. Inclusion and Exclusion Criteria

Our search considered studies that focused on human and animal schistosomiasis. We extracted relevant information and stratified for short-, mid-, and long-term effects of the WBLP. Specifically, we focused on primary data (e.g., the number of people infected with *S. japonicum* or snail infection rates).

Regarding disease management, we considered multi-sectoral coordination, community participation, disease surveillance, high-level supervision, target management, and cost-effectiveness. With regard to technological innovation, we were interested in novel control interventions and disease control strategies. In terms of epidemiology, we focused on studies that documented changes in infection rates in humans, livestock, and *Oncomelania* snails. Articles were excluded that did not assess specific epidemiological or economic aspects attributed to the WBLP.

## 3. Results

As shown in Figure 1, our search revealed a total of 563 articles. However, 456 articles were excluded because they did not address the WBLP for schistosomiasis control in the People’s Republic of China. Reasons for exclusion were (i) WBLP was only referred to in the background without any salient data provided (*n* = 211); (ii) implementing new approaches different from the ones emphasized during WBLP (*n* = 160); and (iii) studies focused on specific populations, such as immigrants (*n* = 87). Another 26 articles were excluded because the studies contained no economic information. Of these, eight articles were excluded as they did not specify costs, and 18 articles were excluded as they mainly tested the extra effect of drug or other means of control interventions. Our final database consisted of 83 articles that address various aspects of the WBLP for schistosomiasis control in the People’s Republic of China (Appendix A). Specifically, 46 articles were related to changes in transmission levels, 22 to technology updates, and 17 to management experience (Figure 2). Most of the included articles evaluated WBLP as an indicator of positive serological tests, positive faecal tests in humans and domestic animals, and areas characterized with a positive snail rate. Two of the 83 articles examined the burden of schistosomiasis japonica, as expressed by disability-adjusted life years (DALYs), and two used an indicator of fatality rate in patients with late-stage schistosomiasis.

### 3.1. Short-Term Effects of the WBLP

The short-term effects of the WBLP investment on schistosomiasis control in the People’s Republic of China are considered to be outcomes of the project during or towards the end of the project. Many reports were published, particularly during the later stage of the project or shortly after completion of the WBLP. It was found that the overall prevalence of *S. japonicum* in humans slightly declined over time from 5.0% in 1992 to 4.7% in 1998. The prevalence rates in the high- and medium-endemic areas decreased by 59.7% (95% confidence interval (CI): 59.3–60.2%) and 26.4% (95% CI: 26.3–26.5%), while it increased by 24.1% (95% CI: 23.9 to 24.3%) in the low-endemic areas. The overall infection rate in livestock (cattle) fell by 62.3% (95% CI: 61.7–62.9%) between 1992 and 1998. Regarding high-, medium-, and low-endemic areas, the respective decreases were 62.3% (95% CI: 60.4–64.2%), 43.9% (95% CI: 40.2–47.4%), and 75.4% (95% CI: 75.3–75.5%). The data confirmed the objectives and pre-set targets of the WBLP were largely met [11].

From 1992 to 1998, snail infection rates in high-, medium-, and low-endemic areas declined by 82.7% (95% CI: 82.5–82.7%), 64.3% (95% CI: 64.1–64.5%), and 38.5% (95% CI: 37.8–39.0%), respectively, and the infection rate of snails at the unit of the county fell by 68.8% (95% CI: 68.6–68.9%). This decline exceeded the pre-set WBLP target of 50–60%. However, considerable heterogeneity in the density of infected snails was observed according to baseline endemicity. In terms of changes in the density of infected snail, reduction rates of 69.0% and 86.5% were observed in the high- and medium-endemic areas, respectively. On the other hand, the density of infected snails increased by 46.5% in low-endemic counties. Importantly though, the areas infested with snails in high-, medium-, and low-endemic areas all expanded to varying degrees; 1.2%, 44.0%, and 21.7%, respectively. The areas infested with snails (excluding Zhejiang province) increased by 14.9% during the WBLP in the 1990s [13].

In 2001, when the WBLP was completed, the total number of human schistosomiasis cases across the People’s Republic of China had dropped to an estimated 820,000 cases. The number of positive livestock declined to an estimated 31,500. The areas with snails shrank from 14,800 km^2^ to 3436 km^2^ [14,15]. There were slight fluctuations in the area with snails over the course of the WBLP. A before and after evaluation of the WBLP project concluded that the objectives of the WBLP were largely achieved.

### 3.2. Mid-Term Effects of the WBLP

Shortly after the termination of the WBLP in 2001, several studies reported a resurgence of schistosomiasis in the Yangtze River drainage basin, including areas that had previously fulfilled the transmission interruption criteria [16,17]. In retrospect, two issues contributed to the resurgence of schistosomiasis along the Yangtze River: (i) the negative impact of a major flooding event in the whole Yangtze River basin in 1998 [18]; and (ii) discontinuity of funding support for schistosomiasis control after cessation of the WBLP. Hence, a mid-term evaluation was conducted, focusing on the period from 1998 to 2003. The analysis concluded that both the prevalence of *S. japonicum* in the human population and the number of acute schistosomiasis cases had rebounded significantly with some acute cases even occurring in urban areas [16,19,20]. No relief in prevention and control was foreseen until schistosomiasis elimination was fully achieved.

This rebound of schistosomiasis was also reflected in antibody levels of the residential population. It was found that the total positivity rate of serological tests increased by 1% from 2002 to 2004 [21]. Of note, the endemic areas showed varying degrees of resurgence, with a more pronounced rise in Yunnan province, probably explained by its unique geography [14]. Positive rates in domestic animals increased by approximately 1%. Areas infected with intermediate host snails also increased slightly. One study explored the effectiveness of the WBLP through changes in the spatial risk distribution of schistosomiasis and showed that MDA was more effective for near-term control, while snail-control measures were less effective [22]. This observation might be explained by reduced susceptibility of *S. japonicum* to praziquantel governed by heavy external dosing, a conclusion supported by earlier studies [8,23]. Hence, research into new drugs for schistosomiasis or reducing resistance to praziquantel is warranted. Another reason might by that the level of community cooperation declined over time.

Detailed analyses revealed five main reasons for the observed increase in schistosomiasis prevalence shortly after termination of the WBLP. First, with the end of the WBLP, specific funding for schistosomiasis control decreased considerably [24]. As schistosomiasis mainly occurs in rural areas where sanitation is poor, interrupting schistosomiasis transmission needs to be supported by water, sanitation, and hygiene (WASH) measures, along with general improvements in health care standards. Second, the effect of snail control was relatively modest [22]. According to WHO guidelines for the prevention and control of schistosomiasis, the main objective of the WBLP was set to reduce morbidity in the human population. Coupled with the fact that snail-control techniques available at the time did not meet the required demand, this did not lead to a significant reduction in the areas infested with snails. Even today, snail control is a difficult endeavour within elimination programmes and requires integration with other measures of environmental management. Third, the availability of praziquantel for MDA, coupled with IEC in the 1990s led to a marked decline in schistosomiasis during the WBLP. However, towards the end of the WBLP and after project completion, adherence of local residents to MDA became more challenging, as residents perceived that oral medication might negatively affect their health with each additional round of treatment. Hence, MDA was gradually scaled down. Fourth, in 1998, severe floods occurred along the Yangtze River, and these damaged large numbers of water construction projects that were aimed at snail control. Additionally, new engineering projects that emphasized anti-flooding measures resulted in the introduction and spread of snails, and hence, snail-infested areas actually increased. Fifth, after the outbreak of severe acute respiratory syndrome (SARS) in 2002, major health resources were diverted from schistosomiasis and other parasitic diseases towards anti-SARS activities. As a result, the efforts on schistosomiasis control declined significantly, when compared with the period before and during the WBLP [25].

Despite a slight increase in the prevalence of schistosomiasis in humans in 2002, the WBLP had established a foundation to guide schistosomiasis control and elimination in. Morbidity due to schistosomiasis was still at a relatively low level, compared with the situation before the WBLP. Furthermore, schistosomiasis transmission was under control in most areas, allowing the government to deploy resources and optimize schistosomiasis control measures after a resurgence occurred in the Yangtze River valley [10].

### 3.3. Long-Term Effects of the WBLP

The reemergence of schistosomiasis at the beginning of the new millennium was a warning sign that, even in low-endemic areas, schistosomiasis still has the potential to rebound if control efforts are reduced or even discontinued. This is mainly explained by the zoonotic nature of the disease (i.e., there are more than 40 animal reservoir hosts) and its complex lifecycle [26]. Hence, only interruption of disease transmission will ensure the health and wellbeing of the local population in endemic areas. The backlash against schistosomiasis caught the attention of the Chinese government, which spurred a multi-sector cooperation mechanism instigated in 2005 [10].

The endemicity of schistosomiasis has declined steadily in the People’s Republic of China since 2005. For instance, the positive rate of serological tests decreased by 5% from 2005 to 2017 [21]. Infection rates of livestock decreased by 3.7%, which was close to zero in 2017. The areas infested with snails in all regions have shown a slight downward trend since 2002, although the decline lacked statistical significance.

Endemic areas for *S. japonicum* have been significantly reduced compared with the situation before the launch of the WBLP. For instance, while schistosomiasis was prevalent in 12 provinces in the People’s Republic of China in 1989, including three types of endemic areas (i.e., water-network region, hilly and mountainous region, and lake and marshland region) [27,28], the areas of endemicity in 2021 were primarily concentrated in the lake and marshland regions of Anhui, Hunan, and Jiangxi provinces. In the other nine provinces, the goal of transmission interruption, and hence, elimination, had been reached.

The burden of schistosomiasis, expressed in DALYs, decreased by 68.7% from 1990 to 2017. Over the same period, the number of deaths attributable to schistosomiasis declined by 79.5% [29]. The prevalence of human schistosomiasis had dropped from 17.3% in 1992 to 2.1% by 2017. The overall decline in schistosomiasis prevalence in the People’s Republic of China has been significant when compared with the baseline situation before the launch of the WBLP project. It is noteworthy to compare the situation with Indonesia. Although the overall trend of schistosomiasis prevalence in Indonesia has also been decreased, the prevalence in the Bada Valley, Central Sulawesi showed a slight increase from 2008 to 2017 [30].

The long-term effect of the WBLP not only resulted in a continuous decline in the density of schistosomiasis transmission, but it also waxed and waned ideas and strategies for project implementation, management, and evaluation. Particularly notable were the following innovations. First, the WBLP validated the feasibility of the WHO guidelines for schistosomiasis elimination in the People’s Republic of China and identified shortcomings in some of the control measures. The WBLP project proved that MDA is feasible for *S. japonicum* control, both in humans and animals, which is a key lesson for schistosomiasis morbidity control and elimination as a public health problem [31]. On the other hand, the WBLP also revealed some shortcomings of MDA, such as the risk of developing drug resistance and a reduction in residents’ compliance to MDA in the face of declining prevalence and intensity of infection coupled with reduced levels of morbidity [32]. These issues call for additional research to improve MDA protocols to provide clear guidance on when tactics must change, so that compliance remains appropriately high.

Second, basic and operational research accompanied the WBLP implementation. Guided by an international scientific and strategic advisory board, that included four experts in epidemiology, health economics, and public health (i.e., David Evans, Bruno Gryseels, Robert Bergquist, and Marcel Tanner), enhanced exposure of Chinese scientists to the global research community. Of particular note was implementation research that was fostered by a Joint Research Management Committee (JRMC) with the participation of Chinese and foreign experts [24,33] (Figure 3). In turn, the WBLP stimulated new ideas and innovation and, hence, schistosomiasis control guided by professionals further increased. Specific training courses were offered to schistosomiasis control staff and project managers to improve their technical and management skills. Taken together, the research capacity of Chinese researchers focusing on the epidemiology and control of schistosomiasis improved considerably in various fields such as research quality, enhanced funding applications, and in monitoring research progress [32].

Third, integrating these valuable experiences, coupled with enhanced capabilities of researchers and disease control managers, paved the way from control to the elimination of schistosomiasis. Concerted efforts in the first two decades of the new millennium, and embracing an integrated, inter-sectoral approach, shows that elimination of schistosomiasis is in reach [33]. Indeed, by 2022, most of the remaining cases of human schistosomiasis are concentrated in lakes and marshland regions that were the most serious areas of endemicity before the launch of the WBLP [34].

Hence, despite the resurgence of schistosomiasis in some parts of the People’s Republic of China shortly after the end of the WBLP, the literature largely affirms that the project provided an evidence-base that schistosomiasis control is feasible, spurring the way towards elimination [35,36]. Indeed, as the prevalence of schistosomiasis gradually decreased, the goal shifted from control to elimination. Although this causes more and more costs per infection avoided, elimination is the declared goal because, once elimination has been achieved, the costs drop, and hence, it is considered a highly cost-effective strategy [37,38].

### 3.4. Effectiveness Evaluation

Regarding cost–benefit analysis studies, the benefits of integrated schistosomiasis control programmes include gains both from reducing the number of infected individuals and in reducing infection rates in intermediate host snails. The former includes treatment benefits, prevention benefits, and workforce benefits, and provides scientific information for allocating input structures and numbers through cost–benefit comparisons. The latter enables the balancing of investments to control programmes with short- or long-term effectiveness [39].

It was found that, by the end of the WBLP, total benefits per capita were USD 1804, with a total benefit–cost ratio of 7.2 and a net benefit–cost ratio of 6.2, for a net benefit value of USD 602.34 million. The correlation coefficients between the net benefit–cost ratio and infection rates of *S. japonicum* in humans and cattle were 0.55 and 0.66, respectively [39], indicating that, the higher transmission intensity in an area, the higher the benefit–cost ratio for the interventions undertaken in the same location. Additionally, the transmission intensity was reduced significantly by the end of 2001, coinciding with the completion of the WBLP. Indeed, the prevalence of schistosomiasis in humans and livestock was reduced by more than 50%, and the density of snail infections at all prevalence levels had fallen by more than 75%. The number of cases of hepatosplenomegaly due to schistosomiasis had also decreased significantly. During the WBLP, Zhejiang province met the national standard for schistosomiasis elimination. Many counties in other provinces met the standard for interrupting transmission [40]. Taken together, the target set at the beginning of the WBLP was reached or even exceeded. These outcomes illustrate that the benefit–cost of the WBLP could be evaluated over a short-term.

An important aspect of this report is that we provide additional evidence on the long-term benefits of the WBLP. We conjecture that the WBLP further accelerated the national schistosomiasis control programme and paved the way from control to elimination. The long-term benefits further improved the working mechanism of the national schistosomiasis control programme, such as improvements in management mechanisms, in surveillance and response systems, and in the development of technical innovations.

The first long-term benefit was an improvement in the management mechanism through community mobilization. In terms of working management, since schistosomiasis posed a significant disease burden from the 1950s to the 1980s, the Chinese government was strongly committed to the control of the disease. The traditional way in management of the national schistosomiasis control programme was to formulate the annual work plans at an all-government level, while control interventions were implemented under government leadership to ensure the smooth implementation of activities [41]. The efforts of schistosomiasis control were carried out nationwide by the central government leadership team, which unified the different central departments and organizations and integrated the resources of the provinces. The core of leadership was devolved to local party committees, with the central government taking overall control. This *modus operandi* was changed during and after the WBLP, as follows. After 1990, the leadership function was transferred to the local governments, which were better placed to coordinate the various sectors and instigate the development of integrated interventions in the implementation of the programme. This change in leadership clarified the tasks at the central and local levels and between local governments and departments. At the same time, a schistosomiasis control system, with clear roles and responsibilities at various levels to smooth operational channels, was set up. Of note, the major duties of the central government focused on financial support of the local control efforts and to provide motivation to control staff. The local governments, in turn, took the initiative in driving the fight against schistosomiasis, as the basis of national administrative action and government governance.

Regarding community mobilization, mass participation played an important role in the early days of the national control programme, such as treatment, prevention, and snail control, since it was to encourage all residents from endemic areas to participate in schistosomiasis control planning under the leadership at all levels of the government [10]. After the WBLP was completed, the preceding channels of participation provided to the public were still available, but more professional staff with experience in schistosomiasis control were involved. As scientific control measures became mainstream, the public played a more cooperative role with the government. Taking IEC as an example, by the end of the WBLP, more than 80% of the population in the schistosomiasis-endemic areas were aware of the objectives of schistosomiasis control. Approximately 70% of residents who could not avoid contact with infested water (e.g., farmers, flood fighters, and boatmen) changed their behaviour and took protective measures (e.g., impregnating clothes with clonidine). Over 90% of the population in the endemic areas cooperated with professional staff in regular screening and treatment activities.

The second long-term benefit of the WBLP was an improved surveillance–response system to eliminate the risk of schistosomiasis transmission [42]. Although it was the most costly part of the national schistosomiasis elimination programme, active and passive surveillance was placed high on the agenda as a core intervention. During the WBLP, a hierarchical set of surveillance measures was adopted, with half of the residents in high-endemic areas being screened annually by faecal examination of those with a prior positive serological result. Additionally, snail infested areas were randomly screened in 40% of the endemic areas to estimate and monitor annual snail infection rates. In low-endemic areas, all children aged 7–14 years were screened every other year, following the same approach as described above, complemented with annual snail surveys in 50% of the areas where snails were present. At the same time, 30% of villages were selected at random in high-endemic areas and utilized as sentinel surveillance sites, while in low- and medium-endemic areas, 1% of villages were selected at random as sentinel surveillance sites. When a township (comprising up to 20 villages) had a low level of prevalence, it was also periodically assessed according to different criteria to determine whether it had reached the criterion of “transmission control” or “transmission interruption” [43].

The third long-term benefit of the WBLP was the improvements in technical innovation through development, validation, and deployment of new strategies and innovative tools for the national schistosomiasis control and elimination programme. Regarding innovation and new strategies in different endemic areas, the following developments are worth mentioning. Firstly, several studies on epidemiological patterns showed that control of the source of schistosomiasis infection prioritized humans over domestic animals [34]. However, other studies indicated that domestic animals (e.g., cattle and goats) play an increasingly important role in the transmission of schistosomiasis [44]. Additionally, wild animals such as deer and giant salamanders, were also found positive for *S. japonicum* infection, and should thus be considered as additional heretofore-unknown hosts for *S. japonicum* transmission [45]. Secondly, the life history characteristics of schistosomes warrant control measures that are multifaceted. Indeed, integrated prevention and control efforts are more cost-effective, and hence, integrated approaches are needed that mobilize resources from the health, agriculture, water, forestry, and education sectors to achieve schistosomiasis prevention and control [24,46]. In 2005, a four-pronged control initiative, including agricultural mechanization, fencing of cattle, improvements in housing facilities (e.g., provision of tap water and improved sanitation), and providing toilets for mobile populations were piloted in four provinces: Jiangxi, Hubei, Hunan, and Sichuan. The objective was to prevent people from polluting the environment with parasite eggs. The results of these pilot studies showed that such a multi-pronged initiative is effective in eliminating schistosomiasis, and even more broadly, has a marked effect in reducing soil-transmitted helminth infections as well [47,48]. Thirdly, MDA with praziquantel in the People’s Republic of China yielded impressive results [42]. Although praziquantel reduces morbidity, several studies concluded that it does little to eliminate infection and that drug resistance might develop should the drug be used indiscriminately [49,50]. Indeed, there is growing consensus that the large-scale use of praziquantel on its own is unlikely to eliminate schistosomiasis [51,52,53]. Fourthly, control strategies should be stratified according to endemicity characterization. For instance, the use of MDA might suffice to control morbidity in high- and medium-endemic areas, but this might need to be complemented with environmental modification to permanently interrupt schistosomiasis transmission in low-endemic areas. Concrete lining of irrigation canals with gates to prevent the spread of snails through irrigation systems might be necessary in other areas. These enhanced control strategies increase effectiveness and pave the way for local elimination. With the national schistosomiasis elimination programme launched in 2015, a stronger surveillance–response system has been articulated with more than 450 surveillance sites set up in the People’s Republic of China, including in all endemic counties [24].

Regarding the development and implementation of advanced technologies for use in schistosomiasis control efforts, the government of the People’s Republic of China placed particular emphasis on the following: the development of new tools for detecting infections both in humans and animals; the prevention of infection through water contact, snail control, and snail elimination; new? drugs to treat infected individuals and to reduce transmission both to humans and animals, among other intervention measures. In particular, during the 10-year WBLP, 245 implementation research projects were granted through the JRMC. To date, 278 articles have been published, 25 research projects were awarded special recognition at the provincial level, and seven research projects were granted national patents. The long-term effectiveness of implementational research through the JRMC greatly contributed to the further advancement of schistosomiasis control and spurred the way for its elimination in the People’s Republic of China.

Several case studies brought to the fore new tools for schistosomiasis control through the JRMC, three of which are highlighted here. A first case study pertains to remote sensing technology, which has been applied to determine risk factors for schistosomiasis transmission [54]. Using remote sensing, transmission patterns of schistosomiasis have been shown to be associated with environmental changes for predicting transmission risks. Natural disasters (e.g., floods) and water resource developments, or environmental changes after construction of the Three Gorges Dam, predicted pathways as to how schistosomiasis is spread, using digital mapping supported by remote sensing data. Transmission risk in different environmental settings, combined with snail habitats, could be predicted, thus guiding control strategies and spatial targeting of surveillance in high-risk areas affected by environmental changes [55].

The second example was the development of facilities to prevent the spread of snails along water courses, based on the results of a study of the hydraulics and biology of *Oncomelania*, the intermediate snail host of *S. japonicum*. This facilitates the building of water courses that are usually handled in corroboration with water conservation engineering projects, and is a cost-effective way to reduce *Oncomelania* snails from spreading [56].

A third example pertains to tools for snail elimination, as this is one of the most effective ways to interrupt the spread of schistosomiasis [51]. However, snail elimination measures are constrained by environmental protection issues and the limited level of funding available [57]. Although the use of chemical molluscicides is quite effective, there are risks associated with environmental changes [58,59]. Moreover, the complex terrain in hilly and mountainous areas in the People’s Republic of China renders snail elimination difficult [60]. Therefore, several new tools, supported by agricultural reform, water conservation, and the planting of snail-proof forests, have proven viable and cost-effective as alternatives to the use of molluscicides [61].

## 4. Lessons Learned

For many years, the global strategy for schistosomiasis control emphasized morbidity control through MDA with praziquantel [62]. The significant reduction in the prevalence and intensity of schistosome infections in humans and livestock and the reduction in snail infection rates during the 10-year WBLP strengthened the evidence that this approach works and might even lead to the elimination of schistosomiasis [63,64]. The results of a cost-effectiveness study of the WBLP show that the 10-year endeavour significantly reduced the prevalence of schistosomiasis in the People’s Republic of China, and had a very substantial protective and promotive effect on people’s health and local socioeconomic development [65].

Schistosomiasis transmission is influenced by environmental changes, including global warming [66,67,68,69]. Some studies have demonstrated that rising temperatures may lead to the spread of schistosome-susceptible snails into new areas. With a projected temperature increase of 0.9 °C by 2030, schistosomiasis transmission could spread from the present southern waterfront to non-endemic areas further north in the People’s Republic of China [70,71]. The global elimination of schistosomiasis by 2030 remains challenging [65,72]. Hence, more effective measures to control and eliminate the disease, including taking into account future environmental changes, need further scientific inquiry. Hookworm infection and other intestinal helminthiases have similar transmission characteristics to schistosomiasis. Hence, the results of this study may provide strategic support and technical standards for the control and even elimination not only of schistosomiasis, but also of soil transmitted helminth infections, and perhaps other neglected tropical diseases as well.

## Figures and Tables

**Figure 1 diseases-10-00084-f001:**
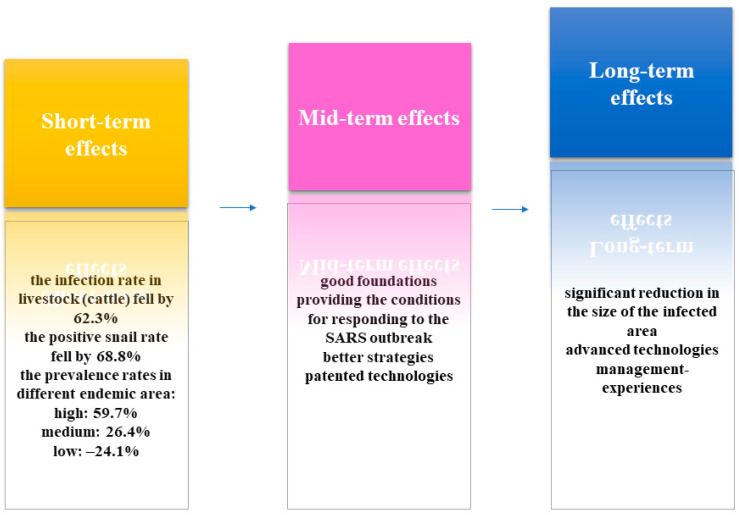
Short-, mid-, and long-term effects of the 10-year World Bank Loan Project (WBLP) for schistosomiasis control in the People’s Republic of China.

**Figure 2 diseases-10-00084-f002:**
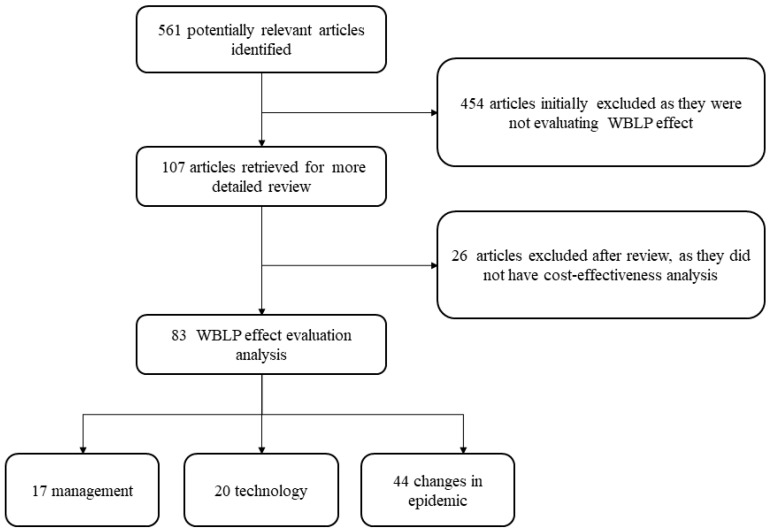
Flow chart to identify studies with relevant information for determining the short-, mid-, and long-term effects of the 10-year World Bank Loan Project (WBLP) for schistosomiasis control in the People’s Republic of China.

**Figure 3 diseases-10-00084-f003:**
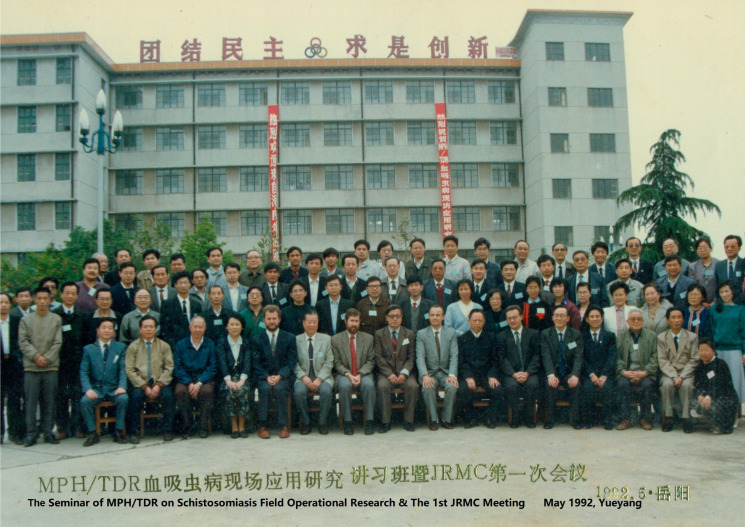
Group photo of initial Joint Research Management Committee (JRMC), held in May 1992 to provide strategic and scientific advice on the 10-year World Bank Loan Project (WBLP) for schistosomiasis control in the People’s Republic of China (Marcel Tanner; first row, 11th from left).

## Data Availability

All the relevant data generated during this study are included in the manuscript.

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
