# Peer review of "Short-, Mid-, and Long-Term Epidemiological and Economic Effects of the World Bank Loan Project on Schistosomiasis Control in the People’s Republic of China"

_diseases, 2022, doi:10.3390/diseases10040084_

Round 1
Reviewer 1 Report
A review of Short-, mid- and long-term epidemiological and economic effects of the World Bank Loan Project on schistosomiasis control in the People’s Republic of China: A few points deserve attention for further publication. I suggest that it is accepted for publication after the following revisions:
- The authors could clarify in the manuscript's abstract the mechanism, advantages, problems, and solutions for the Short-, mid- and long-term epidemiological and economic effects of the World Bank Loan Project on schistosomiasis control in the People’s Republic of China.
- In addition, the authors should highlight the advantages/disadvantages of this Short-, mid- and long-term epidemiological and economic effects of the World Bank Loan Project on schistosomiasis control in the People’s Republic of China methods for industrial application and how this information will be addressed in the manuscript.
- Advantages for Short-, mid- and long-term epidemiological and economic effects of the World Bank Loan Project on schistosomiasis control in the People’s Republic of China systems: Which methods have advantages? Are they simple methods of contribution? When compared with other sustainable techniques? Authors must leave this clear information throughout the text and the methods discussed in this manuscript. In addition, this information is needed for the Short-, mid- and long-term epidemiological and economic effects of the World Bank Loan Project on schistosomiasis control in the People’s Republic of China systems contribution protocols are applied on an industrial scale.
- Problems with Short-, mid- and long-term epidemiological and economic effects of the World Bank Loan Project on schistosomiasis control in the People’s Republic of China systems: Does this protocol have a significant problem? This discussion could be improved.
- Additionally, advances in the Short-, mid- and long-term epidemiological and economic effects of the World Bank Loan Project on schistosomiasis control in the People’s Republic of China systems with engineered tailor-made have been performed with other strategies. May open new opportunities. This discussion could be improved.
- This manuscript has broached interest in the progress and recent applications of Short-, mid- and long-term epidemiological and economic effects of the World Bank Loan Project on schistosomiasis control in the People’s Republic of China: The main contributions to the accomplishment of this work must be included in the conclusion.
- Please check all references according to the author's instructions.
- The manuscript must be formatted according to the journal's standards
Author Response
Reviewer # 1
A review of Short-, mid- and long-term epidemiological and economic effects of the World Bank Loan Project on schistosomiasis control in the People’s Republic of China: A few points deserve attention for further publication. I suggest that it is accepted for publication after the following revisions:
- The authors could clarify in the manuscript's abstract the mechanism, advantages, problems, and solutions for the Short-, mid- and long-term epidemiological and economic effects of the World Bank Loan Project on schistosomiasis control in the People’s Republic of China.
Reply: These points have been addressed and the Abstract amended accordingly (see revised manuscript, lines 24-32).
- In addition, the authors should highlight the advantages/disadvantages of this Short-, mid- and long-term epidemiological and economic effects of the World Bank Loan Project on schistosomiasis control in the People’s Republic of China methods for industrial application and how this information will be addressed in the manuscript.
Reply: We have addressed this issue in the Methods section (see revised manuscript, lines 269-270).
- Advantages for Short-, mid- and long-term epidemiological and economic effects of the World Bank Loan Project on schistosomiasis control in the People’s Republic of China systems: Which methods have advantages? Are they simple methods of contribution? When compared with other sustainable techniques? Authors must leave this clear information throughout the text and the methods discussed in this manuscript. In addition, this information is needed for the Short-, mid- and long-term epidemiological and economic effects of the World Bank Loan Project on schistosomiasis control in the People’s Republic of China systems contribution protocols are applied on an industrial scale.
Reply: The economic evaluation section of our piece presents the experience of the different government-led interventions set up during the WBLP, readily tailored for different settings in the People’s Republic of China. Specifically, we emphasize surveillance and integrated prevention and control.
- Problems with Short-, mid- and long-term epidemiological and economic effects of the World Bank Loan Project on schistosomiasis control in the People’s Republic of China systems: Does this protocol have a significant problem? This discussion could be improved.
Reply: As revealed through our literature review, many articles studying the impact of the WBLP found positive effects of the project in terms of prevention and control of schistosomiasis in the People’s Republic of China. In our view, it is useful to have all relevant articles collected, examined and key findings synthesized, so that lessons learned might guide the prevention and control of schistosomiasis elsewhere.
- Additionally, advances in the Short-, mid- and long-term epidemiological and economic effects of the World Bank Loan Project on schistosomiasis control in the People’s Republic of China systems with engineered tailor-made have been performed with other strategies. May open new opportunities. This discussion could be improved.
Reply: The WBLP made important contributions to international exchange, management and technological innovation, and training of professionals to enhance schistosomiasis control and elimination. These issues are now highlighted more prominently (see revised manuscript, lines 274-277).
- This manuscript has broached interest in the progress and recent applications of Short-, mid- and long-term epidemiological and economic effects of the World Bank Loan Project on schistosomiasis control in the People’s Republic of China: The main contributions to the accomplishment of this work must be included in the conclusion.
Reply: These points have been addressed (see revised manuscript, lines 314-334).
- Please check all references according to the author's instructions.
- The manuscript must be formatted according to the journal's standards
Reply: While revising our piece, we carefully checked for formatting issues, including references.

Reviewer 2 Report
This paper deals with the results obtained from the World Bank loan Project (WBLP) to China on the control of infection with Schistosoma japonicum in humans.
The title should mention S. japonicum, since there are several schistosomiases throughout the world. S. japonicum is excreting eggs in faeces (like S. mansoni) and causes severe liver cirrhosis in humans; it is also a zoonosis since it can be found in cattle and many other animals.
The authors have reviewed available literature from 1992 to 2022, i.e. from the onset to 10 years post-project. They based their paper on 81 articles meeting their criteria in relation to WBLP although they first found 561 articles related to S. japonicum infection. I wonder if this large rejection of paper because they did not include ALL the criteria (such as economic impact for example) is not a loss of information or even a distortion of information. It could have provided more accurate data on prevalence along the 20 years in humans or possibly in animals or infection of intermediate host, the snail Oncomelania. I suggest that the authors comment on this point or justify their choice by showing that for prevalence for example there was no distortion of prevalence.
These 81 articles are providing information on short, medium and long term and in low – medium and highly endemic zones. There are then 6 categories so it means that probably there will be at best 15 reports for each one. The reductions in prevalence are not given with any confidence interval or at least with a range of values; they might be due to chance. I propose that range should be provided on every reduction value.
I would like to know on how many papers and which professions were concerned with the DALYs and the human mortalities DUE to S. japonicum.
Mass treatment for so many years could lead to resistance to praziquantel. Any information on this? Were the cattle also treated since farmers working with them could become easily reinfected? If they were not, it could then have been a refugia and thus maintains susceptibility of Schistosoma.
The paper is concentrated on People’s Republic of China. It could have been nice to get some information on what has been happening in countries without the WBLP (Japan, Sri Lanka or Indonesia..).
I recommend major revision.
Author Response
Reviewer # 2
This paper deals with the results obtained from the World Bank loan Project (WBLP) to China on the control of infection with Schistosoma japonicum in humans.
The title should mention S. japonicum, since there are several schistosomiases throughout the world. S. japonicum is excreting eggs in faeces (like S. mansoni) and causes severe liver cirrhosis in humans; it is also a zoonosis since it can be found in cattle and many other animals.
Reply: The process of preventing and controlling schistosomiasis in the People’s Republic of China is well established. The endemic areas are distributed over different topographies with different socio-economic levels, and hence, they experience different cost inputs for control. The aim of this paper is to derive generic experience in the prevention and control of poverty-related infectious diseases with schistosomiasis as an example in case.
The authors have reviewed available literature from 1992 to 2022, i.e. from the onset to 10 years post-project. They based their paper on 81 articles meeting their criteria in relation to WBLP although they first found 561 articles related to S. japonicum infection. I wonder if this large rejection of paper because they did not include ALL the criteria (such as economic impact for example) is not a loss of information or even a distortion of information. It could have provided more accurate data on prevalence along the 20 years in humans or possibly in animals or infection of intermediate host, the snail Oncomelania. I suggest that the authors comment on this point or justify their choice by showing that for prevalence for example there was no distortion of prevalence.
Reply: We pursued a systematic review with clearly defined inclusion and exclusion criteria. While revising our piece, more details are given regarding the flow chart (see revised manuscript, lines 107-110,123-142; Figure 1).
These 81 articles are providing information on short, medium and long term and in low – medium and highly endemic zones. There are then 6 categories so it means that probably there will be at best 15 reports for each one. The reductions in prevalence are not given with any confidence interval or at least with a range of values; they might be due to chance. I propose that range should be provided on every reduction value.
Reply: This point has been addressed and the manuscript revised accordingly (see revised manuscript, lines 148-160).
I would like to know on how many papers and which professions were concerned with the DALYs and the human mortalities DUE to S. japonicum.[1]
Reply: The requested information is now provided (see revised manuscript, lines 140-142).
Mass treatment for so many years could lead to resistance to praziquantel. Any information on this? Were the cattle also treated since farmers working with them could become easily reinfected? If they were not, it could then have been a refugia and thus maintains susceptibility of Schistosoma.
Reply: Although there is no clinically relevant resistance to praziquantel thus far, the development and spread of such resistance is a concern. We slightly expanded our piece to highlight this issue (see revised manuscript, lines193-200,408-412).
The paper is concentrated on People’s Republic of China. It could have been nice to get some information on what has been happening in countries without the WBLP (Japan, Sri Lanka or Indonesia..).
Reply: Though the focus of our piece is on the People’s Republic of China, we now draw the readers’ attention to some other schistosome-endemic settings (see revised manuscript, lines 53-57,257-262).

Round 2
Reviewer 2 Report
Most of the modifications required are now included in the manuscript. It can be accepted as is.